# Evaluation of Aptamers as Affinity Reagents for an Enhancement of SRM-Based Detection of Low-Abundance Proteins in Blood Plasma

**DOI:** 10.3390/biomedicines8050133

**Published:** 2020-05-24

**Authors:** Sergey Radko, Konstantin Ptitsyn, Svetlana Novikova, Yana Kiseleva, Alexander Moysa, Leonid Kurbatov, Maria Mannanova, Victor Zgoda, Elena Ponomarenko, Andrey Lisitsa, Alexander Archakov

**Affiliations:** 1Institute of Biomedical Chemistry, Moscow 119121, Russia; konstantin157@yandex.ru (K.P.); svetlankin8787@mail.ru (S.N.); a.alexandrmoysa@gmail.com (A.M.); kurbatovl@mail.ru (L.K.); mannanova_m@mail.ru (M.M.); victor.zgoda@gmail.com (V.Z.); elena.ponomarenko@ibmc.msk.ru (E.P.); lisitsa062@gmail.com (A.L.); inst@ibmc.msk.ru (A.A.); 2Russian Scientific Center of Roentgenoradiology, Moscow 117485, Russia; yana.kiseleva@gmail.com

**Keywords:** selected reaction monitoring, protein detection, blood plasma, SMAD4, aptamer-based affinity enrichment

## Abstract

Selected reaction monitoring (SRM) is a mass spectrometric technique characterized by the exceptionally high selectivity and sensitivity of protein detection. However, even with this technique, the quantitative detection of low- and ultralow-abundance proteins in blood plasma, which is of great importance for the search and verification of novel protein disease markers, is a challenging task due to the immense dynamic range of protein abundance levels. One approach used to overcome this problem is the immunoaffinity enrichment of target proteins for SRM analysis, employing monoclonal antibodies. Aptamers appear as a promising alternative to antibodies for affinity enrichment. Here, using recombinant protein SMAD4 as a model target added at known concentrations to human blood plasma and SRM as a detection method, we investigated a relationship between the initial amount of the target protein and its amount in the fraction enriched with SMAD4 by an anti-SMAD4 DNA-aptamer immobilized on magnetic beads. It was found that the aptamer-based enrichment provided a 30-fold increase in the sensitivity of SRM detection of SMAD4. These results indicate that the aptamer-based affinity enrichment of target proteins can be successfully employed to improve quantitative detection of low-abundance proteins by SRM in undepleted human blood plasma.

## 1. Introduction

Mass spectrometry (MS) has today become an indispensable tool for protein quantitative detection and identification in systems biology, biomedical research, and clinical proteomics [1]. However, its application to the detection of proteins present at low and ultralow concentrations in blood plasma or serum, which is of great importance for the search and verification of novel protein disease markers, is limited by the immense dynamic range of protein abundance levels that can span 12 orders of magnitude [2,3,4,5]. While the concentration of high-abundance proteins ranges in plasma from micrograms to tens of milligrams per milliliter, the upper concentration boundary for low-abundance proteins (LAPs) is usually set as 100 ng/mL [6,7,8]. LAPs include tissue-leakage proteins, which are of the most interest as putative biomarkers [9]. For example, the well-known plasma tumor markers are LAPs present at concentrations well below 100 ng/mL [10]. To enhance the sensitivity of MS detection, the plasma (serum) protein dynamic range can be reduced prior to MS analysis either by the removal of the most abundant proteins by means of immunoaffinity columns (immunoaffinity depletion) [11] or by the immunoaffinity enrichment of target proteins with monoclonal antibodies (mAbs) immobilized on a solid phase carrier [12,13]. As an alternative to antibodies, affinity-based reagents, such as DNA and RNA aptamers, have been suggested for a selective enrichment of target proteins [14]. Aptamers have a number of technological advantages over antibodies. Exhibiting (at average) affinities to proteins comparable to those of antibodies, aptamers show no batch-to-batch variability in affinities to a target and are easily immobilized onto various solid phase supports without a loss of affinity via different chemical linkers, which can be attached to the 3'- or 5'-ends of aptamer oligonucleotides during chemical synthesis [15].

The utility of aptamer-based affinity enrichment of targeted proteins for the purpose of MS analysis was shown in a number of publications [16,17,18,19,20,21,22,23,24,25,26]. It is worth noting that the use of aptamers instead of mAbs was reported to result in a remarkable reduction of the complexity of tryptic hydrolysates due to the absence of peptides originating from mAbs [17,22,23]. A coupling of the aptamer-based affinity enrichment of protein targets with different modes of MS was demonstrated, including liquid chromatography–mass spectrometry [23] and liquid chromatography–tandem mass spectrometry [17,18,21,24,26], as well as matrix-assisted laser desorption/ionization–time of flight–mass spectrometry (MALDI-ToF-MS) [16,19,20,22,25]. Among the MS modes used, a particular method known as ‘selected reaction monitoring’ (SRM) is characterized by the exceptionally high selectivity and sensitivity of protein detection [27], allowing the detection of proteins in cell lysates as low as in a few copies per cell for selected molecules [28]. When coupled with immunoaffinity depletion, SRM is able to detect proteins down to picomolar concentrations in blood plasma [4]. However, immunodepletion can result in a complete or partial loss of target proteins due to either their non-specific binding with resin and mAbs or their specific interaction with the removed proteins [29]. While it has been shown that the detection sensitivity can benefit from the coupling of aptamer-based affinity enrichment with SRM for the detection of target proteins in cell lysates [18,24], the question of how the amount of target protein in the enriched fraction, measured by SRM, relates to its initial amount in blood plasma in a wide range of protein concentrations is not answered yet.

In the present work, using recombinant protein SMAD4 (mothers against decapentaplegic homolog 4, the neXtProt accession number NX_Q13485) as a model target added at known concentrations to undepleted human blood plasma samples, we investigated the relationship between the initial amount of the target protein and its amount in the enriched fraction, as determined by SRM. The DNA aptamer constructed on the basis of the consensus sequence recognized by SMAD4 [30] and immobilized on magnetic beads was used for the aptamer-based enrichment of the target protein. It was found that aptamer-based enrichment can enhance the sensitivity of SRM detection of recombinant SMAD4 (rSMAD4) by a factor of 18 to 26, depending on the protein concentration in blood plasma, that results in about a 30-fold decrease of the limit of detection.

## 2. Materials and Methods

### 2.1. Expression and Purification of Recombinant SMAD4

The recombinant SMAD4 protein containing the N-terminal His6-tag was expressed in competent *E. coli* cells of the strain C41 transformed with plasmid pET23a. The plasmid had an insert encoding the SMAD4 protein. Cells of the transformed strain were sonicated, and the cell lysate was cleared by centrifugation (4 °C, 10,000× *g*, 40 min). rSMAD4 was purified using CL-4B Sepharose columns (Merck #CL4B200) saturated with cobalt ions. After passing the lysate through the column, bound proteins were eluted with an imidazole gradient. The maximum elution of rSMAD4 was observed in the range of imidazole concentrations of 60–80 mM. Expression of rSMAD4 was confirmed by the results of electrophoretic and MALDI-ToF-MS analyses. Protein concentrations were measured with the BCA assay (Thermo Fisher Scientific #23235). More details on the expression, purification, and characterization of rSMAD4 may be found in [30].

### 2.2. Synthesis and Characterization of Anti-SMAD4 DNA Aptamer

The anti-SMAD4 DNA aptamer (Table 1) was synthesized in-house on an ASM-800 DNA synthesizer (Biosset, Novosibirsk, Russia) in accordance with the manufacturer’s protocols, using phosphoramidites and CPG from Glen Research (USA). The synthesized oligonucleotides were purified on Glen-Pak purification cartridges (Glen Research #60-5000-96), following the manufacturer’s guidelines. Other chemicals used for oligonucleotide synthesis and purification were from Acros Organics (Belgium). The biotin and FAM (6-carboxyfluorescein) molecules were attached to the 3’-end of the oligonucleotide during chemical synthesis. Purified oligonucleotides were vacuum dried and dissolved in Milli-Q water. Aptamer concentration was determined spectrophotometrically based on the calculated molar extinction coefficient of 4.46·10^5^ and 4.67·10^5^ M^-1^cm^-1^ for biotin and FAM-labeled oligonucleotides, respectively (http://biotools.nubic.northwestern.edu/OligoCalc.html).

The DNA aptamer affinity to rSMAD4 was determined by microscale thermophoresis using a Monolith NT.115 instrument (NanoTemper, München, Germany). The aptamer/rSMAD4 binding was probed in buffer containing 20 mM Tris-HCl (pH 7.4), 5 mM KCl, and 500 mM NaCl. The aptamer concentration in aptamer/rSMAD4 mixtures was 100 nM while the rSMAD4 concentration varied from 0.15 nM up to 2.3 µM. The apparent dissociation constant for aptamer/rSMAD4 complexes was provided by the instrument software, assuming a 1:1 binding stoichiometry.

### 2.3. Aptamer Immobilization on Magnetic Beads

Dynabeads MyOne Streptavidin C1 magnetic beads (Thermo Fisher Scientific #65001) coated with streptavidin were washed twice with 10 mM Tris-HCl, 150 mM NaCl, pH 7.4 (further referred to as buffer B for simplicity), using a DynaMag magnetic separation rack (Thermo Fisher Scientific #12321D). Afterwards, the beads were resuspended in buffer B at the solid concentration of 10 mg/mL. The solution of the anti-SMAD4 aptamer in water was mixed with a concentrate of buffer B to provide a 1 µM aptamer solution in buffer B, incubated at 95 °C for 5 min, and quickly cooled down by placing on ice. The aptamer solution and the suspension of beads were mixed in the ratio of 25 pmoles of the aptamer per 100 µg of solids. The concentration of sodium ions in the mixture was adjusted to 1 M with a concentrated solution of NaCl and the mixture was incubated for 15 min at room temperature (RT) under constant stirring. Beads with attached anti-SMAD4 aptamers were washed twice and resuspended in buffer B at the solid concentration of 10 mg/mL.

### 2.4. Pull-Down of rSMAD4 from Blood Plasma

The samples of human blood plasma collected earlier in the framework of the study on proteomic profiling of blood plasma of healthy individuals [4] were used. Venous blood was collected from volunteers at the Institute of Medico-Biological Problems of the Russian Academy of Sciences (Moscow, Russia) in accordance with the guidelines of the local ethical committees and with the informed consent from all participants [4]. Collected plasma samples were stored at −80 °C. The protein concentration in pooled plasma, measured with the BCA assay, was 73 mg/mL. After thawing, aliquots of plasma were spiked with various amounts of rSMAD4 and 3 µL were sampled from each aliquot to prepare reference samples for a mass spectrometric control of the amount of rSMAD4 in a given aliquot. In total, 27 µL of 2% sodium dodecyl sulfate in 10 mM Tris-HCl, pH 7.5 (buffer L) were added to 3 µL of the sampled plasma and solutions were incubated at 95 °C for 5 min. The rest of each aliquot was used to pull-down rSMAD4 from plasma. For this, 100 µL of plasma were diluted twice with buffer B additionally supplemented with 0.85 M NaCl. Beads carrying anti-SMAD4 aptamers were pelleted using a magnetic rack and the pellet was resuspended in the diluted plasma (75 µL of the bead suspension per 200 µL of the diluted plasma). After 30 min of incubation at RT with constant shaking, beads were pelleted with a magnet, washed once with buffer B additionally supplemented with 0.35 M NaCl, and bound proteins were eluted by resuspending beads in 30 µL of buffer L, followed by incubation at 95 °C for 5 min. Afterwards, the beads were immediately pelleted on the magnetic rack and the eluate was withdrawn.

### 2.5. ‘In Silico’ Selection of Proteotypic Peptides for SMAD4 and Synthesis of Internal Standards

Proteotypic tryptic peptides for the SMAD4 protein were selected in silico as described earlier [4,31] using the online bioinformatics tool ‘Peptide unicity checker’ (https://www.nextprot.org/viewers/unicity-checker/app/index.html). The peptide selection was conducted in compliance with the Human Proteome Project Mass Spectrometry Data Interpretation Guidelines [32]. In particular, peptides must be 8 or more amino acid residues long to ensure the uniqueness of the peptides within a biological species and must not have chemically labile amino acids, such as Cys and Met, in the peptide sequence. Two selected peptides, GWGPDYPR and IYPSAYIK, were synthesized on an automatic peptide synthesizer Overture (Protein Technologies, Tucson, AZ, USA) in accordance with the protocol described elsewhere [33,34]. Their isotope-labeled analogues were also synthesized to serve as internal standards for the quantitative measurement of the amount of rSMAD4 in spiked plasma and pull-downs. The “heavy” peptides were synthesized using isotope-labeled arginine (GWGPDYPR) and lysine (IYPSAYIK) precursors. The concentrations of the synthesized peptides were measured by amino acid analysis with fluorescent signal detection [35] as described earlier [4].

### 2.6. Tryptic Digestion

Eluted proteins and proteins in the corresponding reference sample were subjected to trypsinolysis on filters, using the FASP (filter-aided sample preparation) method [36,37]. The protocol described in [37] was followed with slight modifications. Briefly, 30 µL of each sample were mixed with 200 µL of 8 M urea/2 M thiourea in 100 mM Tris-HCl, pH 8.5 (buffer U) and ultrafiltrated using Microcon centrifugal filter units (30 kDa molecular weight cut-off; Merck #MRCF0R030) at 7500× *g* for 25 min at RT. The filtrate was discarded, 100 µL of buffer U were added into the filtration unit, and the unit was centrifuged again. Then, 50 µL of the freshly prepared 50 mM solution of 2-iodoacetamide in buffer U were pipetted into the unit and incubated in a darkness for 1 h. The filter was washed twice with 100 µL of buffer U, followed by two washes with 100 µL of 50 mM triethylammonium bicarbonate (pH 8.5) buffer (buffer D). For protein digestion, 50 µL of buffer D containing 1.5 µg of trypsin (Promega #V5280) were pipetted into the unit and proteins were digested overnight at 37 °C, followed by the addition of 5 µL of trypsin solution (200 ng/µL), and incubation for an additional 3 h. Then, the isotope-labeled synthetic peptides (internal standards) were spiked into tryptic digests. Afterwards, peptides were collected by centrifugation at 7500× *g* for 25 min at RT and filters were washed twice with 100 µL of 0.1% formic acid in 50% acetonitrile. For each sample, the filtrates were pooled, divided into two parts, and vacuum dried. One part was used for MS analysis and another to estimate the total amount of peptides in samples by the BCA assay. For the latter, peptides were dissolved in 3 M urea with sonication.

### 2.7. SRM Analysis and Data Processing

For SRM analysis, peptides were dissolved in 0.1% formic acid. SRM analysis was carried out using a chromatographic system Dionex UltiMate 3000 RSLCnano System, coupled with a triple quadrupole mass spectrometer TSQ Vantage (Thermo Scientific, Waltham, MA, USA). The analysis was performed in three technical replicates as previously described [34,38]. Briefly, a sample containing 0.7–1 µg of the total peptide was applied onto a Zorbax 300SB-C18 precolumn (Agilent Technologies, Santa Clara, CA, USA) and washed with 5% acetonitrile for 5 min at a flow rate of 10 µL/min before separation on the analytical column. Peptides were separated using the analytical column, Zorbax 300SB-C18 (3.5 µm, 150 mm × 75 µm) (Agilent Technologies, USA), with a linear gradient from 95% solvent A (0.1% formic acid) and 5 % solvent B (80% acetonitrile, 0.1% formic acid) to 60% solvent A and 40% solvent B over 25 min at a flow rate of 0.4 µL/min. The capillary voltage for the electrospray ion source of TSQ Vantage was set at 2100 V, the isolation window was set to 0.7 Da for the first and the third quadrupole, and the cycle time was 3 s. Fragmentation of the precursor ions was performed at 1.0 mTorr, using collision energies calculated by Skyline 3.6.0 software (MacCoss Lab Software, USA) (https://skyline.ms/project/home/software/Skyline/begin.view).

Quantitative analysis of SRM data was performed using Skyline 3.6.0 software. Quantification data were obtained from the "total ratio" numbers calculated by Skyline. Isotopically labeled peptide counterparts were added in known amounts of 25–35 fmole per 1 µg of the total peptide. The results were inspected using Skyline software to compare the chromatographic profiles of endogenous and stable-isotope-labeled peptides. The coefficient of variation of the transition intensity did not exceed 20% in technical runs.

## 3. Results and Discussion

SMAD4 is a common partner for proteins of the RSMAD (receptor-regulated SMADs) group involved in the formation of transcriptionally active complexes induced by transforming growth factor beta [39]. Since SMAD4 is considered as a potential target for targeted cancer therapies [39], we recently tried to select DNA aptamers against recombinant SMAD4 protein (rSMAD4) by the SELEX (Systematic Evolution of Ligands by EXponential enrichment) method [30]. As a result, oligonucleotides containing 5'-GTCT-3’ and 5’-AGAC-3' sequences known as SMAD-binding elements (SBE) have evolved by the end of selection, thus indicating that the SMAD4–SBE interaction dominated the aptamer selection process. The SBE sequences were used to construct the anti-SMAD4 aptamer [30], as shown in Table 1. The equilibrium dissociation constant, *K*_d_, of the aptamer/SMAD4 complexes, measured by microscale thermophoresis, was found to equal (45 ± 7) nM.

Though SMAD4 is not considered as a potential circulating biomarker for cancer diagnostics, this is an intracellular protein that is not normally present in blood plasma at a detectable level and therefore will not interfere with the detection of rSMAD4 spiked in plasma. Moreover, the immobilization of the anti-SMAD4 aptamer via the biotin moiety (Table 1) on streptavidin-coated magnetic beads was shown to allow the selective pull down of rSMAD4 from *E. coli* cell lysates as demonstrated by electrophoretic analysis [30]. Thus, rSMAD4 and the anti-SMAD4 aptamer were employed here as a convenient model for evaluating aptamers as affinity reagents for the enhancement of SRM-based detection of low-abundance proteins in blood plasma.

In SRM, the mass spectrometric analysis is carried out on a triple quadrupole mass spectrometer in which the first mass analyzer is set to filtering mode with a narrow isolation window (with a particular value of mass-to-charge ratio, m/z, and a width, as a rule, of ±1) to isolate a specific peptide (precursor ion) of interest. Then, the isolated peptide is fragmented, and one of the resulting fragment ions is monitored with another mass analyzer set to filter a certain mass-to-charge ratio. The double selection of a peptide precursor ion and a ‘daughter’ fragment ion is called a ‘transition’ and determines the high specificity of the SRM method, resulting in low chemical noise and, consequently, high sensitivity of the protein detection in complex samples [27]. Peptides whose m/z ratios together with m/z ratios for daughter fragments uniquely identify a given protein are called ‘proteotypic peptides’ [40]. For the absolute quantification of a target protein, the pertinent proteotyping peptides are synthesized using stable isotope-labeled amino acid precursors and spiked at known concentrations into analyzed samples as internal standards [41].

Two peptides, GWGPDYPR and IYPSAYIK, were selected in silico as proteotypic for the SMAD4 protein, using bioinformatics tools, and chemically synthesized. Their isotope-labeled analogues were also chemically synthesized to serve as internal standards. In Table 2, a set of corresponding transitions for both native and isotope-labeled (“heavy”) synthetic peptides is shown. Figure 1 presents the SRM spectra for these peptides. As seen, the relations between the signal intensities for the daughter fragment ions differ for the peptides: The domination of y6 ions in a spectrum is much more pronounced for IYPSAYIK, compared to GWGPDYPR, that makes the later peptide look more promising for confident SRM detection. However, when these two peptides were used for the SRM detection of rSMAD4 spiked into plasma, GWGPDYPR turned out to have a worse signal-to-noise ratio and a much greater variability in the detected amount of the protein at low rSMAD4 concentrations, compared to those for IYPSAYIK (data not shown). Apparently, a strong interference from a particular proteolytic matrix peptide (or peptides) is the reason behind the poor performance of GWGPDYPR as a proteotypic peptide for rSMAD4 detection. In contrast, no interference from matrix peptides was observed for peptide IYPSAYIK down to nanomolar concentrations (Figure 2). Consequently, peptide IYPSAYIK and its “heavy” analogue were employed to quantify spiked rSMAD4 in samples of undepleted human plasma in the wide range of rSMAD4 concentrations, from approximately 40 pM up to about 30 µM, and in samples derived from the tested plasma by the anti-SMAD4 aptamer-based enrichment (further referred to as ‘pull-downs’ for simplicity).

Figure 3 shows the molar amount of peptide IYPSAYIK per 1 µg of the total tryptic peptides, measured by SRM in plasma samples and pull-downs, as a function of its expected molar amount calculated based on the known molar concentrations of rSMAD4 and the weight concentration of the total plasma protein (measured as 73 g/L). To calculate the expected molar amount of peptide IYPSAYIK per 1 µg of the total peptide, we assumed that the ratio of the SMAD4 molar concentration to the weight concentration of the total plasma protein holds throughout all sample preparation procedures. As seen from Figure 3 (curve 1), the measured amount of peptide IYPSAYIK is practically directly proportional to that expected from the SMAD4 concentration in plasma within the concentration range of ≈ 10 nM to ≈ 30 µM. Indeed, the linear regression analysis gives in this case the following equation: Y = (0.799 ± 0.005)·X + (0.7 ± 0.8), r = 0.999. The slope is not equal to unity, probably due to differences in the estimation of the protein amount by the BSA assay and the SRM quantification with an internal standard. Below ≈ 10 nM, the measured amount starts to deviate from direct proportionality to the expected values, which may be attributed to an increasing interference from matrix peptides. Finally, the dependence flattens out at SMAD4 concentrations of 40 pM to 1 nM since the signal becomes indistinguishable from the background noise. For the pull-downs, the corresponding dependence (curve 2 in Figure 3) is shifted towards lower SMAD4 concentrations due to the aptamer-based enrichment of the target protein. The dependence can be approximated by two linear functions with different slopes for the concentration ranges from 1 nM up to ≈100 nM and from 1 nM down to ≈40 pM, *viz.* Y = (18.5 ± 0.8)·X + (0.8 ± 0.5), r = 0.996, and Y = (26.0 ± 2.6)·X + (0.09 ± 0.02), r = 0.990. Thus, the aptamer-based enrichment allowed for about a 20-fold increase in the concentration of the target protein over the concentration range tested. The obtained data also clearly demonstrate that the coupling of SRM with the aptamer-based enrichment of SMAD4 allows one to reliably detect this target protein at concentrations that become ‘undetectable’ for the SRM method applied directly to undepleted plasma samples.

To estimate the analytical performance of SRM alone and SRM coupled with the aptamer-based enrichment, we compared the limits of detection (LOD) for both cases. By definition, LOD can be determined as follows [42]: LOD = LOB + 1.645·SD, where LOB is a so-called ‘limit of blank’ and SD is a standard deviation for the measured values of the analyte. LOB is the highest *apparent* analyte concentration expected to be found when replicates of a blank sample containing no analyte are tested. LOB can be calculated as LOB = **mean**_blank_ + 1.645·SD_blank_ [42]. We took the corresponding averages from four experimental points on the flat section of the curve 1 in Figure 3 (background noise) as **mean**_blank_ and SD_blank_ that gives 86 amole (of the measured amount of peptide IYPSAYIK) as an estimate for LOB in our case. For the direct application of SRM to plasma, this corresponds to 2.2 nM of SMAD4 in plasma (curve 1 in Figure 3) while for SRM coupled with the aptamer-based enrichment, 44 pM. To estimate the SD values, we averaged the standard deviations from two experimental points on each curve in Figure 3, which lies right above the LOB. This gave us 6 nM and 0.2 nM of the SMAD4 concentration as estimates for the LOD in the case of SRM directly applied to plasma and coupled with the aptamer-based enrichment, respectively. Thus, the aptamer-based enrichment of SMAD4 provides a 30-fold analytical improvement for SRM detection of this target protein in undepleted plasma. Taking into account that the molecular weight of SMAD4 is around 60 kDa, LODs for its direct and aptamer-assisted SRM detection can be calculated as weight concentrations of 12 and 360 ng/mL. The LOD of 12 ng/mL matches the requirement for LAP detection, in contrast to that for SRM applied directly to plasma.

It appears interesting to compare the LOD achieved with the aptamer-based enrichment to LODs for SRM detection coupled with an antibody-based enrichment. For instance, the antibody-based enrichment of carcinoembryonic antigen allowed an LOD of 15 ng/mL [43] to be achieved while for cardiac troponin, the antibody-based enrichment allowed its quantification down to 11 ng/mL [44]. Recently, Säll et al. [45] reported LODs of 0.05 to 12 ng/mL for 11 proteins detected in undepleted plasma by SRM coupled with the antibody-assisted enrichment. They used the oriented immobilization of different recombinant single-chain antibodies on magnetic beads. Apparently, an LOD for the affinity-based enhancement of SRM detection depends on a range of experimental parameters, including the affinity of a particular antibody (or aptamer) to a given target protein. The anti-SMAD4 aptamer characterized by the *K*_d_ value of 45 nM provides the LOD of 12 ng/mL, which is in general comparable with LODs for the antibody-assisted SRM detection [43,44,45]. It is thought that the use of more potent aptamers might further decrease the LOD of the aptamer-assisted SRM detection.

In summary, we demonstrated that aptamers immobilized on a solid carrier, such as magnetic beads, can be successfully employed as affinity reagents for an enhancement of SRM-based detection of LAPs directly in blood plasma. For rSMAD4 and the anti-SMAD4 DNA-aptamer constructed based on the SBE sequence, the LOD is decreased by a factor of 30. The growing number of aptamers developed for human proteins of diagnostic interest [46] and improvements in aptamer selection methodologies, including the use of modified nucleobases to increase aptamer affinity and selectivity [47,48], make the aptamer-based affinity enrichment of target proteins for the purpose of mass spectrometric analysis a very promising alternative to that based on antibodies.

## Figures and Tables

**Figure 1 biomedicines-08-00133-f001:**
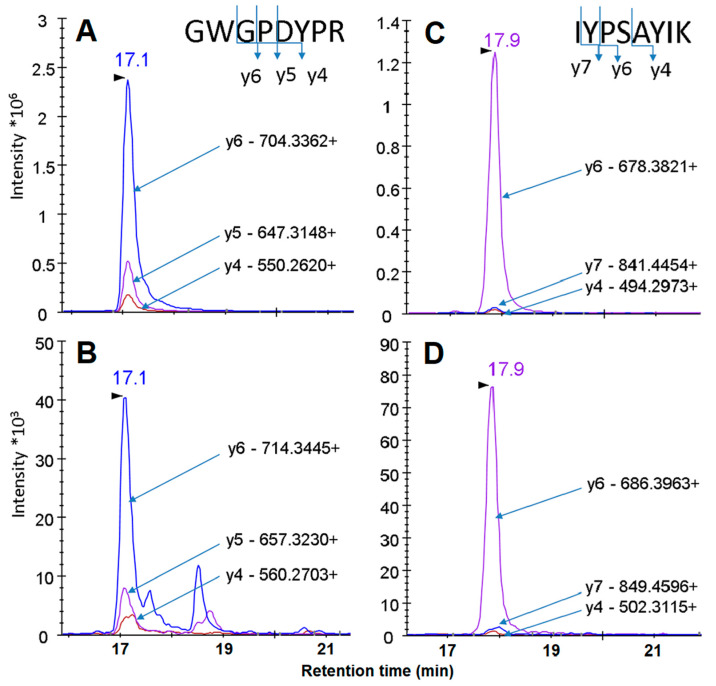
SRM spectra of proteotypic peptides GWGPDYPR (panels **A** and **B**) and IYPSAYIK (panel **C** and **D**) of SMAD4. Panels **A** and **C**: native peptides; panel **B** and **D**: isotope-labeled peptides (internal standards). Spectra were visualized with Skyline 3.6.0 software. The retention time for the major peak indicated by the arrow is shown above the peak.

**Figure 2 biomedicines-08-00133-f002:**
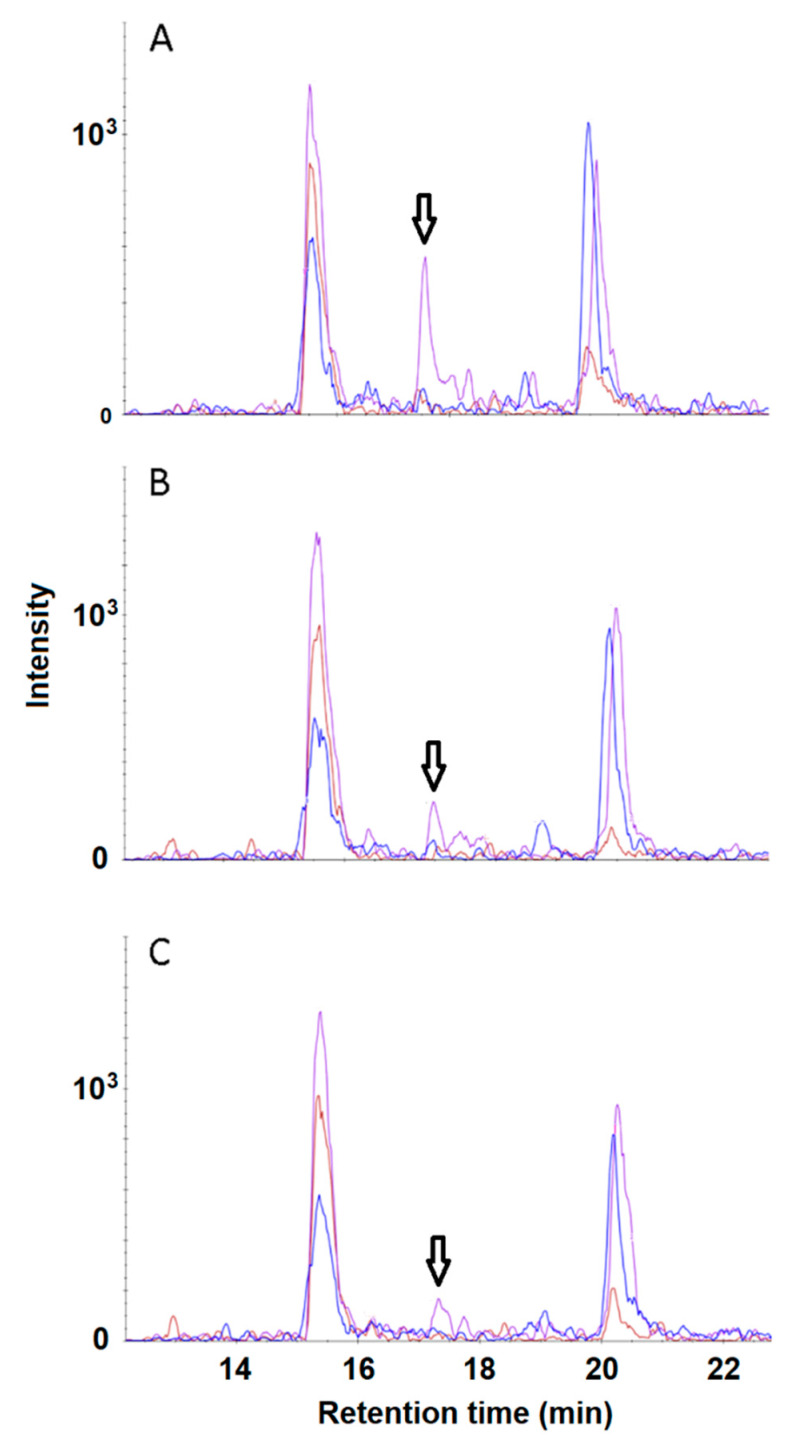
SRM spectra of proteotypic peptide IYPSAYIK of SMAD4 for samples derived from human plasma. The most intensive transition (y6 ‘daughter’ ion) is shown by the arrow. The concentrations of SMAD4 spiked in plasma are 50 nM (**A**), 10 nM (**B**), and 5 nM (**C**). Spectra were visualized with Skyline 3.6.0 software.

**Figure 3 biomedicines-08-00133-f003:**
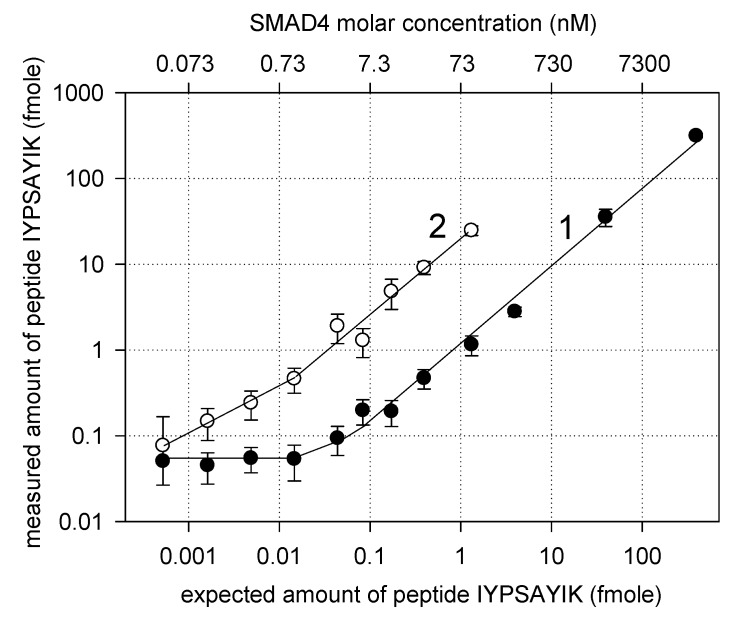
Dependence of the amount of peptide IYPSAYIK measured by SRM on its expected amount. Curve 1: SRM analysis of samples directly derived from plasma spiked with SMAD4; curve 2: SRM analysis of samples derived from plasma with the aptamer-based enrichment of the SMAD4 protein. The expected amount of peptide IYPSAYIK is calculated for samples derived directly from plasma. The upper axis shows the SMAD4 concentration in plasma. The mean values and corresponding standard deviations from three measurements are shown.

**Table 1 biomedicines-08-00133-t001:** Sequence of DNA-aptamer to rSMAD4 protein. The complementary parts of the sequence are underlined. The regions responsible for binding to SMAD4 (SMAD-binding element, SBE) and poly(dA)_5_ linker are shown in bold and italic, respectively; ‘M’ stands for biotin or FAM (6-carboxyfluorescein) attached to the 3’-terminus.

Aptamer Name	Aptamer Sequence
Anti-SMAD4	5'-CGAAGTCTAGACAGCGTTTTCGCTGTCTAGACTTCG*AAAAA*-3'-M

**Table 2 biomedicines-08-00133-t002:** List of transitions for the selected proteotypic peptides of SMAD4. The charge of the precursor and fragment ions is given in superscript.

Peptide Sequences	m/z of Precursor Ion	m/z of Fragment Ion	Collision Energy, eV	Ion Type
GWGPDYPR	474.222 ^+2^	704.336 ^+ 1^	17.1	y6
		647.315 ^+ 1^		y5
		550.262 ^+ 1^		y4
	479.226 ^+2^	714.344 ^+ 1^		y6 (“heavy”)
		657.323 ^+ 1^		y5 (“heavy”)
		560.270 ^+ 1^		y4 (“heavy”)
IYPSAYIK	477.768 ^+2^	841.4454 ^+ 1^	17.2	y7
		678.3821 ^+ 1^		y6
		494.2973 ^+ 1^		y4
	481.775 ^+2^	849.4596 ^+ 1^		y7 (“heavy”)
		686.3963 ^+ 1^		y6 (“heavy”)
		502.3115 ^+ 1^		y4 (“heavy”)

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
