# Peer review of "Evaluation of Aptamers as Affinity Reagents for an Enhancement of SRM-Based Detection of Low-Abundance Proteins in Blood Plasma"

_biomedicines, 2020, doi:10.3390/biomedicines8050133_

Round 1

Reviewer 1 Report

In their manuscript, Radko et al present a proof-of-context study on the use of aptamers as affinity reagents for the enhancement of SMR-based detection of low abundant proteins in blood plasma. The authors show that their assay allows to detect the test protein SMAD4 down to a LOD of 44 pM resulting in a 30-fold analytical improvement over a non-enriched sample.  

As such they reach the goal of this study, which is to test whether protein-specific aptamers can be used for low-abundance protein enrichment from blood plasma samples. While aptamers have been used for protein enrichment from cell lysates, the same procedure had not been tested yet for blood plasma proteins.

Several points should to be further discussed to improve the quality of the manuscript.

 Relevance of protein detection in blood plasma: Line 38 briefly mentions the relevance of this research but a more detailed paragraph with examples would be helpful to illustrate the importance of low-abundant protein detection in blood plasma.

 The proof-of-concept protein SMAD4: It does not get clear to me why this protein was chosen? Lines 78-80 briefly mention its role as potential cancer target, but it should be explained in more detail why knowing the SMAD4 concentration in blood plasma is important and what are potential applications and next steps for the assay developed herein.

 The employed aptamer: The employed aptamer was isolated in a different study. Although the study is cited, it would be helpful to mention the affinity of this aptamer for SMAD4 and discuss how this affinity might change given the difference in buffer conditions used in this study.

LOD: The LOD of the herein tested assay was found to be 44 pM. This value should be put into the context of expected concentrations of low-abundant biomarker proteins. Is the current detection limit enough to detect these proteins? If not, how could the process be improved?  Higher affinity aptamers?

Stability of aptamers in blood plasma: What is known about nucleases in human serum and are their concerns about aptamer stability in blood plasma?

Context of analytical improvement factor should be discussed. The presented assay allows for a 30-fold analytical improvement due to protein enrichment. How does that compare to improvement factors reached with antibody-based enrichment and aptamer-based enrichment in different body fluids?

Other details:

Figure 2 is wrongly labeled as Figure 3

Author Response

Response to Reviewer 1

Relevance of protein detection in blood plasma: Line 38 briefly mentions the relevance of this research but a more detailed paragraph with examples would be helpful to illustrate the importance of low-abundant protein detection in blood plasma.

We have enlarged the pertinent discussion (lines 40 to 44 of the revised manuscript) and added relevant references (refs. 9 and 10) as suggested by the reviewer.

The proof-of-concept protein SMAD4: It does not get clear to me why this protein was chosen? Lines 78-80 briefly mention its role as potential cancer target, but it should be explained in more detail why knowing the SMAD4 concentration in blood plasma is important and what are potential applications and next steps for the assay developed herein.

SMAD4 was chosen as a model target since normally this protein should not be present in blood and therefore will not interfere with the detection of target protein spiked in blood plasma. Thus, as a model protein, SMAD4 has no direct relevance to cancer diagnostics in the context of the present study. We have tried to clarify this point upon revision (lines 92-99 of the revised manuscript) to address the reviewer’s concern.

The employed aptamer: The employed aptamer was isolated in a different study. Although the study is cited, it would be helpful to mention the affinity of this aptamer for SMAD4 and discuss how this affinity might change given the difference in buffer conditions used in this study.

The pull-downs were carried out in plasma diluted twice with 0.85 M NaCl in order to suppress unspecific electrostatic interactions of the aptamer with plasma proteins. The expected sodium concentration should be around 500 mM. The dissociation constant for the rSMAD4/aptamer complex, measured in the buffer with a close sodium concentration (20 mM Tris-HCl, pH 7.4, 5 mM KCl, and 500 mM NaCl) by microscale thermophoresis, was found to equal 45 nM. We provided this estimate and the relevant experimental section in the revised manuscript (lines 90-91 and 239-244) as suggested by the reviewer.

LOD: The LOD of the herein tested assay was found to be 44 pM. This value should be put into the context of expected concentrations of low-abundant biomarker proteins. Is the current detection limit enough to detect these proteins? If not, how could the process be improved? Higher affinity aptamers?

The LOB (limit of blank) was equal to 44 pM. LOB is highest apparent analyte concentration expected to be found when replicates of a blank sample containing no analyte are tested. The LOD was 200 pM that corresponds to 12 ng/mL (the molecular weight of rSNAD4 is around 60 kDa). The upper concentration boundary for low abundance proteins is commonly set as 100 ng/mL (e.g., refs. 6-8). Thus, the LOD for aptamer-assisted SMR detection appears to match the requirement for low abundant protein detection, in contrast to SMR applied directly to plasma – in this case LOD was 6 nM or about 360 ng/mL. The pertinent discussion has been added upon revision (lines 40-44 and 192-195) to address the reviewer comment.

Stability of aptamers in blood plasma: What is known about nucleases in human serum and are their concerns about aptamer stability in blood plasma?

Undoubtedly, the cleavage of aptamers by plasma nucleases is one of bottlenecks for therapeutic applications of aptamers. However, it appears that for the majority of diagnostic applications the aptamer degradation by nucleases is not a major issue, likely due to relatively short incubation times. However, some decrease in the number of “active” aptamers due to nuclease cleavage cannot be ruled out both in our case and in general.

Context of analytical improvement factor should be discussed. The presented assay allows for a 30-fold analytical improvement due to protein enrichment. How does that compare to improvement factors reached with antibody-based enrichment and aptamer-based enrichment in different body fluids?

Unfortunately, in publications where the antibody-based enrichment was employed to enhance the sensitivity of SRM detection of low abundant proteins in plasma, no LODs for detection without the enrichment are provided. Thus, the analytical improvement factor cannot be calculated. To address the reviewer remark, we compared LOD achieved in our study with those reported in the literature. For instance, the antibody-based enrichment of carcinoembryonic antigen (CEA) allowed to achieve LOD of 15 ng/mL (ref. 35) while for cardiac troponin the antibody-based enrichment allowed its quantification down to 11 ng/mL (ref. 36). Apparently, the analytical improvement factor depends on a range of experimental parameters, including affinity of a particular antibody to a given target protein. For example, Sall et al. (ref. 37) reported LODs of 0.05 to 12 ng/mL for 11 proteins tested when they coupled SMR to the antibody-based enrichment with recombinant single chain antibodies allowing oriented immobilization to magnetic beads. Thus, the LOD provided by the aptamer-based enrichment coupled to SMR has a comparable level with those provided by the antibody-based enrichment. The relevant discussion has been added upon revision (lines 196-207 of the revised manuscript).

Other details:

Figure 2 is wrongly labeled as Figure 3

The mistake has been corrected.

All changes made upon revision are highlighted with yellow.

Reviewer 2 Report

The manuscript “Evaluation of aptamers as affinity reagents for an enhancement of SMR-based detection of low abundant proteins in blood plasma” submitted by Sergey Radko and coworkers proposes the application of Single Reaction Monitoring (SRM) MS following purification of the target protein by using specific aptamers immobilized on magnetic beads. In particular, the authors used recombinant SMAD4 to supplement the blood plasma obtained from patients in order to evaluate the response of the method. The authors claim that the use of aptamer-based affinity enrichment can be successfully employed to improve quantitative detection of proteins from human plasma.  Overall, the paper is well-written and potentially interesting for the field. However, the novelty of the present work is low since SRM together with aptamer-based enrichment of the target proteins has been previously described for other target proteins.

Major concerns:

  • Line 102. The authors mention ” Two peptides, GWGPDYPR and IYPSAYIK, were selected in silico as proteotypic for the SMAD4”. Please explain the criteria used for the selection of such peptides. Why the authors did not selected more peptides for their analysis? A combined determination of multiple peptides of SMAD4 by SRM would provide a more accurate determination of SMAD4 levels. Please explain this possibility.

  • A linear regression analysis of the data should be provided in Figure 3.

  • Further evidence to demonstrate the specific binding of the aptamer to the SMAD4 should be provided. I would suggest the authors to provide IP with SDS-PAGE and wb analyses to show the binding and specific recovery of SMAD4.

Minor comments:

  • Please revise numeration of Figures. Figure 3 appears twice in the manuscript.

  • Revise description of the abbreviations in the text. SOB was defined as limit of detection but it does not appear in the text.

Author Response

Response to Reviewer 2

However, the novelty of the present work is low since SRM together with aptamer-based enrichment of the target proteins has been previously described for other target proteins.

The SMR together with aptamer-based enrichment of target proteins was indeed reported earlier (refs. 18 and 24). However, we are not aware of any publications devoted to the aptamer-assisted SMR detection of low abundant proteins in blood plasma. The blood plasma is a specific object due to the immense dynamic range of protein abundance levels.

Major concerns:

Line 102. The authors mention ” Two peptides, GWGPDYPR and IYPSAYIK, were selected in silico as proteotypic for the SMAD4”. Please explain the criteria used for the selection of such peptides. Why the authors did not selected more peptides for their analysis? A combined determination of multiple peptides of SMAD4 by SRM would provide a more accurate determination of SMAD4 levels. Please explain this possibility.

To conduct SRM assay, we selected two isotope labeled peptides to serve as internal standards. The sequence of each of these peptides uniquely characterizes the SMAD4 protein. The rules of selection of the proteotypic sequences of tryptic peptides were described previously (refs. 4 and 41) and are compliant with the recommendations published in 2016 (ref. 42). The main parameters for the choice of proteotypic peptides were the peptide length (8 or more amino acid residues) to ensure the uniqueness of the peptides within a biological species and the absence in the peptide sequence of chemically labile amino acids (Cys and Met). We have provided more details on peptide selection upon revision (lines ?? to ??) to address the reviewer concern.

Two peptides are recommended and commonly used for the reliable protein identification and quantitation in shotgun proteomics (ref. 42). However, in targeted proteomics using SRM and data-independent acquisition it is often enough to have one peptide for the reliable protein identification and quantitation (e.g., refs. 35 and 41 of the manuscript, Emirbayer et al 2017 DOI: 10.1002/pmic.201600455, to mention a few).

A linear regression analysis of the data should be provided in Figure 3.

The linear regression analysis has been conducted as suggested by the reviewer. The results of this analysis are presented and discussed in the revised manuscript (lines 154-157).

Further evidence to demonstrate the specific binding of the aptamer to the SMAD4 should be provided. I would suggest the authors to provide IP with SDS-PAGE and wb analyses to show the binding and specific recovery of SMAD4.

The selective pull-down of rSMAD4 by the anti-SMAD4 aptamer immobilized on magnetic beads was carried out earlier from cell lysates by the protocol identical to that used in the present study, followed by gel electrophoresis analysis (ref. 30). The specifically mentioned it in the revised manuscript (lines 94-97).

Minor comments:

Please revise numeration of Figures. Figure 3 appears twice in the manuscript.

The figure numbering has been corrected.

Revise description of the abbreviations in the text. SOB was defined as limit of detection but it does not appear in the text.

            Abbreviations have been checked and corrected.

All changes made upon revision are highlighted with yellow.

Round 2

Reviewer 2 Report

After revision the authors solved al the main issues present in the first version of the manuscript. In my opinion the work can be accepted for publication in present form.